# A Game Theoretic Analysis of Additive Adversarial Attacks and Defenses

**Ambar Pal**
Mathematical Institute for Data Science
Johns Hopkins University
ambar@jhu.edu

**René Vidal**
Mathematical Institute for Data Science
Johns Hopkins University
rvidal@jhu.edu

## Abstract

Research in adversarial learning follows a cat and mouse game between attackers and defenders where attacks are proposed, they are mitigated by new defenses, and subsequently new attacks are proposed that break earlier defenses, and so on. However, it has remained unclear as to whether there are conditions under which no better attacks or defenses can be proposed. In this paper, we propose a game-theoretic framework for studying attacks and defenses which exist in equilibrium. Under a locally linear decision boundary model for the underlying binary classifier, we prove that the Fast Gradient Method attack and a Randomized Smoothing defense form a Nash Equilibrium. We then show how this equilibrium defense can be approximated given finitely many samples from a data-generating distribution, and derive a generalization bound for the performance of our approximation.

## 1 Introduction

Neural network classifiers have been shown to be vulnerable to additive perturbations to the input, which can cause an anomalous change in the classification output. There are several *attack* methods to compute such perturbations for any input instance which assume access to the model gradient information, e.g., Fast Gradient Sign Method [12] and Projected Gradient Method [24]. In response to such additive attacks, researchers have proposed many additive *defense* methods with varying levels of success, e.g., Randomized Smoothing [7]. However, it has been later discovered that a lot of these defenses are in turn susceptible to further additive attacks handcrafted for the particular defenses. This back and forth where attacks are proposed breaking previous defenses and then further defenses are proposed mitigating earlier attacks has been going on for some time in the community and there are several open questions to be answered. Can all defenses be broken, or do there exist defenses for which we can get provable guarantees? Similarly, does there always exist a defense against any attack, or are there attacks with provable performance degradation guarantees? Do there exist scenarios under which attackers always win, and similarly scenarios where defenders always win? Are there conditions under which an equilibrium between attacks and defenses exist?

In this work, we answer some of these questions in the affirmative in a binary classification setting with locally linear decision boundaries. Specifically, we find a pair of attack $A$ and defense $D$ such that if the attacker uses $A$, then there is no defense which can perform better than $D$, and vice versa. Our approach can be seen as a novel way to obtain both provable attacks and provable defenses, complementing recent advances on certifiable defenses in the literature ([37, 4, 15, 9, 35, 14, 19, 22]).

To summarize, our contributions are as follows:

1. We introduce a game-theoretic framework for studying equilibria of attacks and defenses on an underlying binary classifier. In order to do so, we first specify the capabilities, i.e., the changes the attacker and defender are allowed to make to the input, the amount of knowledge the attacker and the defender have about each other, and formalize the strategies that they can follow.

2. We show that the Fast Gradient Method attack and a Randomized Smoothing defense [1] form a Nash Equilibrium under the assumption of a zero-sum game with locally linear decision boundary for the underlying binary classifier and full knowledge of the data-generating distribution.

3. We propose an optimization-based method to approximate the optimal defense given access to a finite training set of $n$ independent samples and we derive generalization bounds on the performance of this finite-sample approximation. Our bounds show that the approximation approaches the optimal defense at a fast rate of $O(\sqrt{\log n / n})$ with the number of samples $n$.

The rest of the paper is organized as follows: In Section 2 we describe the proposed game theoretic setup. In Section 3 we state our main result showing the existence of an optimal attack and defense. This is followed by Section 4 where we propose an optimization method to approximate the optimal defense, and provide some experiments validating our methods and models. Section 4 presents a generalization analysis showing that the proposed approximation approaches the optimal defense at a fast rate. Finally we conclude in Section 6 by putting our work into perspective with related work.

## 2 A Game Theoretic Setup for Additive Adversarial Attacks and Defenses

We will denote a random variable with an upper-case letter, e.g., $X$, and a realization of a random variable with a lower-case letter, e.g., $x$. We will consider a binary classification task with data distribution $p_X$ defined over the input space $\mathcal{X} \subset \mathbb{R}^m$. The true decision boundary corresponding to the discrete labels $\{-1, +1\}$ will be defined by the zero-contour of a classifier $f: \mathcal{X} \to \mathbb{R}$, i.e., $\{x: f(x) = 0\}$, and the label of each data point $x \in \mathcal{X}$ will be given by $\mathrm{sgn}(f(x))$.

We will define a two-player, single-shot, simultaneous, zero-sum game between an attacker $A$ and a defender $D$. In this setting the attacker and defender are allowed to simultaneously make additive perturbations $a(x)$ and $d(x)$, respectively, for a given a data point $x$, i.e., both $A$ and $D$ submit their perturbation at the same time and the perturbed point is $x + a(x) + d(x)$. The size of each perturbation is limited to be at most $\epsilon$ in $\ell_2$ norm, i.e., for all $x \in \mathcal{X}$, $a(x), d(x) \in V$, where $V := \{v: \|v\|_2 \leq \epsilon\}$.

The score $u_A$ assigned to $A$ is determined by whether the label of the data point $x$ has changed under a locally linear approximation of $f$ around $x$ after both the perturbations $a(x)$ and $d(x)$ are applied:

$$u_A(x, a(x), d(x)) = \begin{cases} +1 & \text{if } \mathrm{sgn}(f_L(x)) \neq \mathrm{sgn}(f_L(x + a(x) + d(x))) \\ -1 & \text{otherwise.} \end{cases} \quad (1)$$

In the above, $f_L(x') = f(x) + \nabla f(x)^\top (x' - x)$ is a linear approximation of $f$ in the $2\epsilon$ neighbourhood of $x$. Similarly, the score $u_D$ assigned to $D$ is defined as the negative of the utility assigned to $A$, i.e., $u_D(x, a, d) = -u_A(x, a, d)$ for all $(x, a, d)$, thus making the game zero-sum.

Note that the locally linear model assumption holds whenever the data distribution places no mass in regions of space that are less than $2\epsilon$ distance away from curved parts of the decision boundary. Concretely, for a neural network with ReLU activations, the assumption holds whenever none of the data points lie very close to the *intersection* of 2 or more hyperplanes which make up the classification boundaries. This is reasonable as such regions form a set of measure zero in the input space. We refer the reader to Sec. E of the Appendix for a more detailed discussion about the validity of such locally-flat boundary assumptions for deep neural networks.

A deterministic strategy for the attacker consists of choosing a function

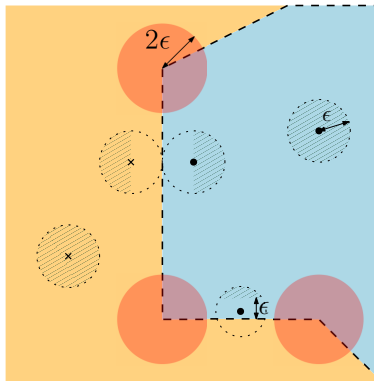

Figure 1: The decision boundary is given by the dashed line. In order for our locally linear modeling assumption to hold, datapoints should lie anywhere except in the red regions, i.e., within $2\epsilon$ distance to a half-plane intersection. The green shaded regions show the geometry of the robust sets $R(x)$, which follows from Lemma 1. Observe that $R(x)$ becomes larger as $x$ moves farther away from the decision boundary.

$a\colon \mathcal{X} \to V$ which dictates the perturbation $a(x)$ that is made by $A$ for the point $x \in \mathcal{X}$. Hence, the action space, i.e., the set of all deterministic strategies that can be followed by the attacker, is the function set $\mathcal{A}_A = \{a | a\colon \mathcal{X} \to V\}$.

A deterministic strategy for the defender consists of the set of all constant functions, i.e., functions which take the same perturbation direction for each point $x \in \mathcal{X}$. Hence the action space $\mathcal{A}_D$ for the defender consists of the function set $\mathcal{A}_D = \{d_v | v \in V, d_v\colon \mathcal{X} \to V \text{ s.t. } \forall x \in \mathcal{X}, \ d_v(x) = v\}$. Since each deterministic strategy for the defender can be uniquely with a point $v \in V$, the reader can think of $\mathcal{A}_D$ as $V$ for ease of understanding. The reasons for constraining the set of strategies for the defender are twofold: first, we want to model existing literature in adversarial attacks where defenders typically follow a single strategy (e.g., smooth input, quantize input) agnostic of the test data point. Second, this restriction captures the fact that defenders are typically given the adversarially perturbed input $x + a(x)$ and are expected to fix it without knowing the original label or original data point $x$ (and hence have to use a strategy that is agnostic to the relation between the data point and its label).

In reality, attackers and defenders can choose even randomized strategies, where the function $a$ or $d$ is sampled according to some probability density on the set of allowed deterministic strategies. Such a randomized strategy $s_A$ for the attacker is specified by a density $p_A \in \mathcal{P}(\mathcal{A}_A)$, where we define $\mathcal{P}(\mathcal{A}_A)$ to be the set of all probability densities over $\mathcal{A}_A$. Similarly, a randomized strategy $s_D$ for the defender is specified by a density $p_D \in \mathcal{P}(\mathcal{A}_D)$. Given a pair of strategies $(s_A, s_D)$, we define the utility function of the attacker, $\bar{u}_A\colon \mathcal{P}(\mathcal{A}_A) \times \mathcal{P}(\mathcal{A}_D) \to \mathbb{R}$ as follows:

$$\bar{u}_A(s_A, s_D) = \mathop{\mathbb{E}}_{x \sim p_X, a \sim p_A, d \sim p_D} u_A(x, a(x), d(x)) \tag{2}$$

Note that since the game is zero-sum, the corresponding utility function $\bar{u}_D$ for the defender is just the negative of that for the attacker. To complete the setup, we specify that both the attacker and defender have *perfect knowledge* about the (possibly randomized) strategy that the other follows, and have access to the linear approximation $f_L$ around any data point $x$. In summary, we have a white-box evasion attack scenario and aim to analyze preprocessing attacks and defenses (i.e. they preprocess the input before they are fed to the classifier) that make additive perturbations to the input.

In the following section, we will further assume both $A$ and $D$ have access to the full data-generating distribution $p_X$ in order to derive the optimal attack and defense strategy. We will show later that this assumption is not needed in practice by constructing an approximation given finitely many samples from $p_X$. We will then prove that this approximation approaches the true optimum at a fast rate.

## 3    A Characterization of Optimal Attack and Defense Strategies

In this section, we will show in Theorem 1 that the FGM attack and Randomized Smoothing defense exist in a Nash Equilibrium, i.e., if $A$ follows the strategy defined by the FGM attack then $D$ cannot do better than following a randomized smoothing strategy, and vice-versa. In order to show this, we will first establish in Lemma 2 that the FGM attack is the best response to any defense in our setting, and then show in Lemma 3 that randomized smoothing is a best response to FGM. Together, these lemmas would lead to our main result. Before stating our results, we begin by defining the robust set.

**Definition 1** (**Robust set**)**.** *The robust set* $R(x) = \{v \colon v \in V \text{ s.t. } \forall v' \in V \ u_A(x, v', v) = -1\}$ *of a point* $x \in \mathcal{X}$ *is the subset of allowed perturbations* $v \in V$ *such that if the defender plays* $v$ *at* $x$*, then the attacker always gets an utility of* $-1$*, i.e., the least possible utility, no matter what she plays.*

Lemma 1 shows that $R(x)$ is the intersection of a half-plane with $V$, as illustrated by the green shaded regions in Fig. 1. Observe that when $x$ is far from the decision boundary, the robust set is equal to $V$.

**Lemma 1** (**Geometry of the robust set**)**.** *For any* $x \in \mathcal{X}$*, the robust set* $R(x)$ *is given by*

$$R(x) = \{v \colon sgn(f(x))(f(x) + \nabla f(x)^\top v) - \epsilon \|\nabla f(x)\| \geq 0\} \cap \{v \colon \|v\|_2 \leq \epsilon\}. \tag{3}$$

Interestingly, the proof of the lemma shows that for a fixed $x$, in order for $v$ to achieve $u_A(x, v', v) = -1$ for all $v' \in V$, it is sufficient to ensure that $u_A(x, v', v) = -1$ when $v'$ is chosen according to the Fast Gradient Method (FGM). The FGM attack was proposed in [12] and makes the additive perturbation $a_{\text{FGM}}(x) = -\epsilon \frac{sgn(f(x))}{\|\nabla f(x)\|_2} \nabla f(x)$. Note that the original attack was called Fast Gradient Sign Method, as it was derived for $\ell_\infty$ bounded perturbations. The same attack for $\ell_2$ bounded perturbations is called the FGM attack. Since this attack does not involve any randomness, the strategy $s_{\text{FGM}}$ followed by the attacker in our framework places probability 1 on the function $a_{\text{FGM}}$.

**Lemma 2 (FGM is a best-response to any defense).** *For any strategy $s_D \in \mathcal{P}(\mathcal{A}_D)$ played by the defender D, the strategy $s_{FGM} \in \mathcal{P}(\mathcal{A}_A)$ played by the attacker A achieves the largest possible utility against $s_D$, i.e., $\bar{u}_A(s_{FGM}, s_D) \geq \bar{u}_A(s_A, s_D)$ for all $s_A \in \mathcal{P}(\mathcal{A}_A)$.*

Having established the optimality of the FGM attack, we now turn our attention to finding an optimal defense strategy. To that end, we define an instance of the Randomized Smoothing defense proposed in [7] that will be shown to be a best-response to FGM. For any perturbation $v \in V$, we define $\phi(v)$ to be the measure of the set of points in $\mathcal{X}$ whose robust sets cointain $v$:

$$\phi(v) = \int_{\mathcal{X}} \mathbf{1}[R(x) \ni v] p_X(x) dx \tag{4}$$

To define $s_{\text{SMOOTH}}$ we need to specify the probability distribution $p_{\text{SMOOTH}} \in \mathcal{P}(\mathcal{A}_D)$. The idea is to sample uniformly from the set of maximizers of $\phi$, i.e., $V^* = \{v^* : \phi(v^*) \geq \phi(v), \; \forall v \in V\}$. Accordingly, our defense strategy $s_{\text{SMOOTH}}$ samples from a uniform distribution over the set of functions $F^* = \{d_{v^*} : v^* \in V^*\} \subseteq \mathcal{A}_D$, where recall $d_v : \mathcal{X} \to V$ is the constant function defined as $\forall x \in \mathcal{X} \; d_v(x) = v$.

**Lemma 3 (Randomized Smoothing is a best-response to FGM).** *The strategy $s_{SMOOTH}$ achieves the largest possible utility for the defender against the attack $s_{FGM}$ played by the attacker, i.e., for any defense $s_D \in \mathcal{P}(\mathcal{A}_D)$ we have $\bar{u}_D(s_{FGM}, s_{SMOOTH}) \geq \bar{u}_D(s_{FGM}, s_D)$.*

Note that Lemma 2 is a stronger result than we need for our further analysis, and we will only be using the following implication of Lemma 2:

**Corollary 2.1 (FGM is a best-response to Randomized Smoothing).** *The strategy $s_{FGM} \in \mathcal{P}(\mathcal{A}_A)$ played by the attacker A achieves the largest possible utility against $s_{SMOOTH}$, i.e., $\bar{u}_A(s_{FGM}, s_{SMOOTH}) \geq \bar{u}_A(s_A, s_{SMOOTH})$ for all $s_A \in \mathcal{P}(\mathcal{A}_A)$.*

As a consequence of Corollary 2.1 and Lemma 3 we have established our main result, as follows:

**Theorem 1 ((FGM, Randomized Smoothing) form a Nash Equilibrium).** *Neither player gains utility by unilaterally deviating when A plays $s_{FGM}$ and D plays $s_{SMOOTH}$, i.e., $\forall s_A \in \mathcal{P}(\mathcal{A}_A), s_D \in \mathcal{P}(\mathcal{A}_D)$, we have $\bar{u}_A(s_{FGM}, s_{SMOOTH}) \geq \bar{u}_A(s_A, s_{SMOOTH})$ and $\bar{u}_D(s_{FGM}, s_{SMOOTH}) \geq \bar{u}_D(s_{FGM}, s_D)$.*

**Implications.** At this point we will pause to note some implications of Theorem 1.

1. *Theoretical insight:* First, Theorem 1 gives us a new theoretical insight into randomized-smoothing. Specifically, one should select the smoothing distribution according to the classifier $f$ to obtain an optimal defense, instead of sampling from the rotationally symmetric Gaussian distribution, which completely ignores the effect of the classifier.

2. *Provable attacks:* Second, Theorem 1 shows that in some settings some attacks are *optimal* in the sense that they will perform better than any alternative regardless of the defense that is employed. This motivates the study of *provable attacks*, something that has been largely ignored by the community which focusses a lot on *provable defenses*.

3. *Winner takes all:* Third, we see from the proofs of Lemmas 2 and 3 that the equilibrium utility obtained by the equilibrium attacker is $1 - 2\phi(v^*)$, which shows that whenever the classification boundaries are such that $\phi(v^*) = 0$, the attacker will always win, i.e., obtain a utility of 1 over the entire dataset. Similarly, the utility obtained by the equilibrium defender is $2\phi(v^*) - 1$, meaning that the defender wins completely whenever the classification boundaries are such that $\phi(v^*) = 1$. Relating this to the widely used metric *robust accuracy*, the above is a characterization of cases where the robust accuracy obtained by the best possible defense is $0\%$ and $100\%$ respectively.

As we saw in this section, we need access to the full data-generating distribution $p_X$ in order to compute $s_{\text{smooth}}$, which is an unreasonable assumption in practice. Hence, in the following section we will demonstrate how one can approximate $s_{\text{smooth}}$ given access to finitely many samples from $p_X$.

## 4 Approximation Properties: How to compute the optimal defense?

As we saw in Section 3, the optimal defense relies on the knowledge of the subset of perturbation directions that maximize $\phi$, $V^*$, which in turns depends on the distribution $p_X$ of the input data.

Given $n$ i.i.d. samples $X_1, X_2, \ldots, X_n \sim p_X$, we define the finite-sample approximation of $\phi$ as:

$$\phi_n(v) = \frac{1}{n} \sum_{i=1}^{n} \mathbf{1}[v \in R(X_i)]. \tag{5}$$

A straightforward modification of the proof of Lemma 3 shows that the same result can be obtained even if $p_{\text{smooth}}$ places all its mass on a single element of $V^*$. That is, an optimal defense can also be achieved by a deterministic strategy. Hence, our goal will now be to solve the following problem:

$$v_n^* = \max_v \phi_n(v) \text{ subject to } \|v\|_2 \le \epsilon. \tag{6}$$

Recall from Lemma 1 that the robust set at $x_i$ can be written as $R(x_i) = \{v \in V : c_i^\top v + b_i \ge 0\}$, where $c_i = \text{sgn}(f(x_i))\nabla f(x_i)$ and $b_i = |f(x_i)| - \epsilon\|\nabla f(x_i)\|$. Therefore, we obtain the following equivalent version of Eq. (6):

$$\max_v \frac{1}{n} \sum_{i=1}^{n} \mathbf{1}[c_i^\top v + b_i \ge 0] \text{ subject to } \|v\|_2 \le \epsilon. \tag{7}$$

Observe that the objective in (7) takes values in $\{0, \frac{1}{n}, \frac{2}{n}, \ldots, 1\}$ and it is equal to 1 iff there is a $v$ with $\|v\| \le \epsilon$ such that $c_i^\top v + b_i \ge 0$ for all $i = 1, \ldots, n$. Trivially, this happens if $b_i \ge 0$ for all $i = 1, \ldots, n$, in which case we can choose $v = 0$, meaning that no defense is needed. By inspection, we see that $b_i \ge 0$ when $|f(x_i)|$ is large, meaning that the classifier is confident about its prediction, and $\|\nabla f(x_i)\|$ is small, meaning that the response of the classifier is not very sensitive to input perturbations. This is very consistent with our intuition that defenses are needed when the classifier is not very confident (small $|f(x_i)|$) or its response is sensitive to input perturbations (large $\|\nabla f(x_i)\|$).

To solve the optimization problem in (7), consider the case where there is a single sample, i.e., $n = 1$. In this case, if the half-space $\mathcal{H} = \{v : c_1^\top v + b_1 \ge 0\}$ does not intersect the hypersphere $B(0, \epsilon)$, then any $v^* \in V$ is a solution to (7) with $\phi(v^*) = 0$. Else, if $\mathcal{H}$ intersects the hypersphere, a solution is given by the projection of the origin onto $\mathcal{H}$, which gives $\phi(v^*) = 1$ (see left panel of Fig. 2 for illustration). When $n = 2$, there are up to two hyperplanes, which divide the space into at most 4 regions. When no half-space intersects the hypersphere, any $v^* \in V$ is an optimal solution and $\phi(v^*) = 0$. When only one half-space intersects the hypersphere, as before an optimal solution is given by the projection of the origin onto the half-space, which gives $\phi(v^*) = 1/2$. When both half-spaces intersect the hypersphere, but the half-spaces intersect each other outside the hypersphere, $v^*$ can be the projection of the origin onto either half-space. It is only when both half-spaces intersect inside the hypersphere we have $\phi(v^*) = 1$ and a maximizer is given by the projection of the origin onto the intersection of both half-spaces (see right panel of Fig. 2 for illustration). However, as $n$ increases, the number of regions grows exponentially in $n$, rendering such a direct region-enumeration intractable. We thus follow an optimization-based approach to find an approximate maximizer of (7). More specifically, we use projected gradient descent on $v$ with the constraint set $\|v\|_2 \le \epsilon$ to solve the optimization problem in (7) and obtain $\widehat{v}_n^*$. Additionally, as the gradients of the indicator function are not very useful, we use the relaxation $\mathbf{1}[\alpha \ge 0] \ge \min(\max(0, \alpha), 1)$ and optimize the RHS.

Our experiments are conducted on the MNIST and FMNIST datasets restricted to two classes. We train a 4-layer convolutional neural network with ReLU activation functions for this binary classification task. The classification results are shown in Table 1, from which we can draw two main conclusions: (1) If the defender uses the equilibrium defense, then the attacker gets the most reduction in approximate accuracy [2] when using the equilibrium attack, as using any other attack improves the performance of the defended classifier. A similar statement holds from the

Table 1: Mean (Variance) of the approximate accuracy computed over binary classifications tasks on MNIST and FMNIST corresponding to $\binom{10}{2}$ pairs of classes.

| Attack | Defense | MNIST (%) | FMNIST (%) |
|--------|---------|-----------|------------|
| - | - | 99.9 (0.0) | 99.9 (0.1) |
| FGM | - | 53.3 (10.0) | 47.4 (5.1) |
| FGM | SMOOTH | 71.2 (14.2) | 67.4 (9.0) |
| PGD | - | 71.9 (12.0) | 74.7 (7.3) |
| PGD | SMOOTH | 94.0 (4.0) | 90.3 (8.5) |

[2] Approximate Accuracy is defined as the accuracy of the model obtained by linearizing the decision boundary in a $2\epsilon$-ball around data-points, thus satisfying our modelling assumption. A detailed description, as well as more experimental details can be found in Sec. D of the Appendix.

other side. (2) The equilibrium defense SMOOTH leads to significant gains in approximate accuracy against both FGM and PGD, in agreement with our result that SMOOTH is optimal when the decision boundaries satisfy our model.

# 5  Generalization Properties of the Approximation to the Optimal Defense

In the previous section, we saw how to practically approximate the optimal defense as $v_n^*$ given access to a finite number of samples drawn i.i.d. from the data distribution $p_X$. But how good is this approximation? In this section, we derive generalization bounds showing that $\phi(v_n^*)$ approaches $\phi(v^*)$ at a fast rate w.r.t. $n$. Before proceeding, we review some necessary results from learning theory.

**Learning theory review.** A function $h\colon \mathbb{R}^m \times \ldots \times \mathbb{R}^m \to \mathbb{R}$ is said to satisfy the *bounded difference* assumption if for all $1 \leq i \leq n$ there exists finite $c_i \in \mathbb{R}$ such that:

$$\sup_{x_1, x_2, \ldots, x_i, \ldots x_n, x_i' \in \mathbb{R}^m} |h(x_1, x_2, \ldots, x_i, \ldots, x_n) - h(x_1, x_2, \ldots, x_i', \ldots, x_n)| \leq c_i. \tag{8}$$

In other words, the bounded difference assumption states that $h$ changes by at most a finite amount if any of the individual inputs are changed, while keeping all others constant. Now, let $h$ be a function satisfying the bounded difference assumption, and $X_1, \ldots, X_n \sim p_X$ be i.i.d. random variables. Then, $h$ satisfies the following useful property called the McDiarmid's inequality, which shows that that the function values of $h$ are tightly concentrated around the mean:

$$\Pr\left[|h(X_1, \ldots, X_n) - \mathbb{E}_{X_1, \ldots, X_n \sim p_X} h(X_1, \ldots, X_n)| > \epsilon\right] \leq \exp\left\{\frac{-2\epsilon^2}{\sum_i c_i^2}\right\}. \tag{9}$$

Next we need some results from *Vapnik-Chervonenkis Theory*. Let $X_1, \ldots, X_n \sim p_X$ be i.i.d. random variables each taking values in $\mathbb{R}^m$. Let $\mathcal{B}$ be a family of subsets of $\mathbb{R}^m$. Let $B \in \mathcal{B}$ be any subset in the family. Define $\mu$ as $\mu(B) = \Pr[X_1 \in B]$. Further, given a particular realization $x_1, \ldots, x_n$ of $X_1, \ldots X_n$, define the finite-sample approximation $\mu_n$ as $\mu_n(B) = \frac{1}{n} \sum_{i=1}^n 1[X_i \in B]$. In other words, $\mu(B)$ is the probability that a sample from $p_X$ lies in $B$, and $\mu_n(B)$ estimates this probability using $n$ samples from $\mathcal{D}$. Taking $h(x_1, \ldots, x_n) = \sup_{B \in \mathcal{B}} |\mu_n(B) - \mu(B)|$ in McDiarmid's inequality, we observe that $c_i = \frac{1}{n}$ and we get the following:

$$\Pr\left[\left|\sup_{B \in \mathcal{B}} |\mu_n(B) - \mu(B)| - \mathbb{E}_{X_1, \ldots, X_n \sim p_X} \sup_{B \in \mathcal{B}} |\mu_n(B) - \mu(B)|\right| > \epsilon\right] \leq \exp\left\{-2n\epsilon^2\right\}. \tag{10}$$

In other words, we see that the maximum inaccuracy incurred in estimating $\Pr[X_1 \in B]$ from finite samples is tightly concentrated around the mean. The final piece we need from VC theory is an upper bound on this mean inaccuracy:

$$\mathbb{E}_{X_1, \ldots, X_n \sim p_X} \sup_{B \in \mathcal{B}} \left|\mu_n(B) - \mu(B)\right| \leq 2\sqrt{\frac{2 \log S_{\mathcal{B}}(n)}{n}}, \tag{11}$$

where $S_{\mathcal{B}}(n)$ is the shatter coefficient for the family $\mathcal{B}$, which is defined as follows:

$$S_{\mathcal{B}}(n) = \sup_{x_1, x_2, \ldots, x_n \in \mathbb{R}^m} \left|\left\{\{x_1, x_2, \ldots, x_n\} \cap B \colon B \in \mathcal{B}\right\}\right|. \tag{12}$$

In the above, each of the terms being considered in the supremum counts the number of distinct intersections with members of $\mathcal{B}$. For illustration, say we are working in $\mathbb{R}^2$, and let $\mathcal{B}$ be the family of subsets generated by taking each rectangle $r$ in the plane and considering $B_r$ to be the points contained in $r$. Now given any 3 points $\{x_1, x_2, x_3\}$ in the plane, we can find rectangles $r$ such that $\{x_1, x_2, x_3\} \cap B_r$ equals each of the 8 possible subsets $\{\}, \{x_1\}, \{x_2\}, \{x_1, x_2\}, \ldots, \{x_1, x_2, x_3\}$. This shows that $S_{\mathcal{B}}(3) \geq 8$ (which implies $S_{\mathcal{B}}(3) = 8$ as $S_{\mathcal{B}}(n) \leq 2^n$).

In other words, the shatter coefficient at $n$ equals the largest $p$ such that $n$ points can be broken into $p$ subsets by members of $\mathcal{B}$. Hence, $S_{\mathcal{B}}(n) \geq p$ implies there exists at least one such example of $x_1, \ldots, x_n$ that can be broken into $p$ subsets. On the other hand, $S_{\mathcal{B}}(n) < p + 1$ implies that *for every* choice of $n$ points, they cannot be broken into $p + 1$ or more distinct sets by members in $\mathcal{B}$.

**Generalization bound.** We will now apply the above literature to our setup. Going forward, we will think of $n$ as being the size of the entire training data that is available to us. Recall that in our

setting, we are given a fixed base classifier $f$. Given $f$, and a sample $x_1, \ldots, x_n$, we know the robust sets $R(x_1), R(x_2), \ldots, R(x_n)$. For each direction that can be taken by the defender, i.e., $v \in V$, we define $B_v$ to be the subset of $\mathcal{X}$ for which $v$ is a robust direction as:

$$B_v = \{x \in \mathcal{X} \colon v \in R(x)\}. \tag{13}$$

Now we can define the family of subsets $\mathcal{B} = \{B_v \colon v \in V\}$. With this definition, we can see that $\phi$ corresponds to $\mu$, and $\phi_n$ corresponds to $\mu_n$ as:

$$\phi(v) := \mu(B_v) = \Pr[X_1 \in B_v] = \Pr[v \in R(X_1)] \tag{14}$$

$$\phi_n(v) := \sum_{i=1}^{n} 1[v \in R(x_i)] = \mu_n(B_v). \tag{15}$$

Given the base classifier $f$ and the training data, we used an optimization algorithm to obtain the maximizer of $\phi_n(v)$ in Section 4. We will assume for the purposes of this section that there is no optimization error, i.e., our optimizer finds the best direction for the given training data, i.e., $v_n^* = \arg\max_v \phi_n(v)$. We are interested in the difference between $\phi(v_n^*)$ and $\phi(v^*)$, i.e.:

$$\phi(v^*) - \phi(v_n^*) = \Big(\phi(v^*) - \phi_n(v_n^*)\Big) + \Big(\phi_n(v_n^*) - \phi(v_n^*)\Big) \tag{16}$$

$$\leq |\phi(v^*) - \phi_n(v^*)| + |\phi_n(v_n^*) - \phi(v_n^*)| \tag{17}$$

$$\leq \sup_v |\phi(v) - \phi_n(v)| + \sup_v |\phi_n(v) - \phi(v)| = 2\sup_v |\phi(v) - \phi_n(v)|. \tag{18}$$

In other words, we can get an upper bound on the quantity of interest by analysing $\sup_v |\phi(v) - \phi_n(v)|$, which is the largest inaccuracy we get due to estimating $\phi(v)$ from a finite number of samples. (10) shows that this quantity is sharply concentrated at its mean, which can be upper bounded by (11) as:

$$\mathbb{E}_{X_1, \ldots, X_n \sim p_X} \sup_{v \in V} \left| \phi_n(v) - \phi(v) \right| \leq 2\sqrt{\frac{2 \log S_{\mathcal{B}}(n)}{n}}. \tag{19}$$

Hence, the problem boils down to getting an upper bound on $S_{\mathcal{B}}(n)$. For families where we can obtain a bound that is sub-exponential in $n$, we can see that the RHS converges to $0$ as $n$ becomes large. Hence, we will now upper-bound the shatter coefficient $S_{\mathcal{B}}(n)$ for our setting.

Recall that we approximate the base classifier $f$ around each data point $x$ using a linear approximation $f_L(x') = f(x) + \nabla f(x)^\top (x' - x)$. We have seen that the robust set is the region enclosed between a half-plane and the boundary of the set $V$ (see Fig. 1 and Lemma 1).

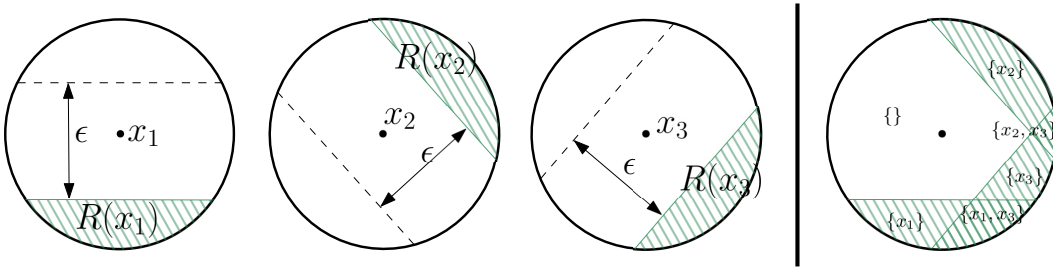

Figure 2: We can compute an upper bound to $S_{\mathcal{B}}$ by looking at the different regions formed by the robust sets $R(x_i)$. The rightmost panel shows the superimposition of $R(x_1), R(x_2), R(x_3)$, showing the different subsets formed.

We now want to upper-bound the maximum number of different partitions of $\{x_1, x_2, \ldots, x_n\}$ that can be formed by taking subsets specified by $B_v$ for $v \in V$. Observe that overlaying all the robust sets in $V$ gives us a collection of regions, with the property that $B_v \cup \{x_1, x_2, \ldots, x_n\}$ is constant when $v$ is varied inside any region, as shown in Fig. 2.

This implies that an upper bound on $S_{\mathcal{B}}(n)$ is equal to the number of distinct regions formed. When we are in the 2-dimensional case, this is same as the number of regions formed by $n$ lines in a plane, which is known to be $(n^2 + n + 2)/2 = O(n^2)$. For higher dimensions $m$, the maximum possible number of regions grows as $O(n^m)$. Thus the upper bound given by (19) reduces to:

$$\mathbb{E}_{X_1, \ldots, X_n \sim p_X} \sup_{v \in V} \left| \phi_n(v) - \phi(v) \right| \leq 4\sqrt{\frac{m \log n}{n}}. \tag{20}$$

The above shows that as $n$ gets larger, i.e., we take more and more samples ($m$ is a constant), we approach the best defense at a fast rate of $O(\sqrt{\log n / n})$. This bound indicates that efficient learning from finite samples is possible. This is corroborated by our experiments, which show even faster rates. We thus believe that our generalization analysis can be sharpened by using recent advances in PAC-Bayesian learning theory, as well as modern extensions to VC-Theory.

## 6    Conclusion, Related Work and Future Directions

In this paper, we have proposed a game theoretic framework under which adversarial attacks and defenses can be studied. Under a locally linear assumption on the decision boundary of the underlying binary classifier, we identified a pair of attack and defense that exist in a Nash Equilibrium in our framework. We then gave an optimization procedure to practically compute the equilibrium defense for any given classifier, and derived generalization bounds for its performance on unseen test data.

There has been a lot of work on the task of classification in the presence of an adversary who can make additive perturbations to the input before passing it to the classifier. There is a huge body of work on empirical attacks in the literature [25, 28, 26, 30, 18, 5, 29, 6, 16, 11, 34], as well as empirically motivated work trying to mitigate the proposed attacks [25, 10, 40, 38, 33]. Here, we will focus on classical game-theoretic approaches to the problem of adversarial classification, as well as other recent defenses for which one can get theoretical guarantees on the performance under attack.

Adversarial classification has been studied in the context of email spam detection, where we have a dataset $(\mathcal{X}, \mathcal{Y})$, and the adversary is allowed to modify the positive (spam) examples in the dataset (i.e., data poisoning attack) by replacing $(x, y)$ by $(x', y)$ where $y = 1$, incurring a cost of modification $c(x, x')$ according to a cost function $c$. The defender is allowed to choose a classifier $h$, which classifies any $x \in \mathcal{X}$ into two classes, i.e., spam or not spam. [8] studied a single shot non-zero sum game with this setup, where the defender always chooses the naïve Bayes classifier given the attacked dataset $(\mathcal{X}', \mathcal{Y})$. [8] set up an integer linear program to compute the best-response of the attacker, and give algorithms to compute the solutions efficiently. [17, 32] have studied similar setups in a sequential setting (called a Stackelberg game), where the attacker goes first and submits the perturbed dataset $\mathcal{X}'$ to the defender, who then learns the classifier having observed $\mathcal{X}'$. [17] showed that the Stackelberg equilibrium can be approximated using optimization techniques, and provide analyses for various classification losses. [41] analyzed the same setting where the classifier is now an SVM classifier, and showed that the $\min \max$ optimization problem arising from the analysis of the Nash Equilibrium of the game can be solved efficiently. [13] approached the problem from a PAC-learning perspective, showing that under certain separability assumptions on the cost function $c$ one can efficiently learn a classifier that attains low error on the attacked training set, as well as maximizes the defender's utility.

A related line of work called adversarial hypothesis testing deals with modifications to the distribution $p_X$ from which the data is sampled instead of modifying the dataset per sample. The utility of the defender now has an additional negative term corresponding to a discrepancy function between the original $p_X$ and the modified distribution $q$. The defender now has to determine whether a given sample of $n$ points came from $p_X$ or $q$, and the defender's utility consists of a tradeoff between the Type-I and Type-II errors in this situation. [39] proved that mixed-strategy Nash equilibria exist in this setting, characterized them, and proved convergence properties of the classification error. [3, 2] study Nash Equilibria for a similar setting, where the defender is now a learner who has to output the weights of a classifier that predicts which distribution the input was sampled from.

Our work differs from past literature as our defender plays an additive perturbation, instead of giving a classifier. In line with practice, we consider the base classifier fixed and provided to us to attack or defend. Additionally, we focus on the case of an attacker that can perform additive perturbations.

Finally, our work links to a recent line of work on certifiable defenses for neural networks (we refer the reader to [7] for a nice review). We focus here on randomized certifiable defenses. [19] gave lower bounds on the robust accuracy of a randomized-smoothing defense via differential-privacy analyses. Subsequently, [23, 7, 21] sharpened the analysis and presented alternative techniques to obtain near-optimal smoothing guarantees for Gaussian smoothed classifiers. Our work complements these proof techniques in the literature as we use geometry of the robust sets as our primary tool to analyze the equilibria, and their optimization and generalization properties.

There are several directions for future work. The first direction would be to extend our results to accommodate locally curved decision boundaries. We suspect that for even further extensions to base classifiers having arbitrarily complex decision boundaries, the style we use to show our generalization results would not yield useful bounds, and we would have to resort to more sophisticated tools to show convergence to the optimal defense. The second direction would be to improve the optimizer of $\phi_n$, and obtain guarantees on the optimization error.

## Broader Impact

At a high level, this work aims to provide a way to characterize adversarial attacks and defenses that might be *best* for each other, in a game theoretic sense where the attacker cannot decrease the robust accuracy further when the defense is fixed, and the defender cannot increase the robust accuracy further when the attack is fixed. The technical contributions are novel geometry-flavored proof techniques that can be used to analyze provable attacks and defenses, and a game-theoretic framework to study such equilibria. Machine learning systems are increasingly being used in security-critical applications, like healthcare and automated driving: our work can be used to find guarantees on the worst accuracy a defended classifier can have under any attack. This is a step towards safe machine learning, where the ultimate goal is to be able to construct classifiers whose performance cannot be degraded by an adversary on most data-points with high probability.

## Acknowledgments and Disclosure of Funding

This work was supported by DARPA Grant HR00112020010 and NSF Grant 1934979.

## Footnotes

[1]Note that our randomized smoothing defense does *not* sample from an isotropic gaussian distribution.

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
