[Supplementary Material]

# A    Proof of Lemma 1

Figure 3: Geometry of the Robust Set $R(x_i)$ is shown by the shaded region. The linear approximation around $x_i$, i.e. $f_L$, is shown by the dotted line. The orange and the blue regions are the two classification regions defined by $f_L$. The perturbation budget is $\epsilon$.

**Lemma 1 (Geometry of the robust set).** *For any $x \in \mathcal{X}$, the robust set $R(x)$ is given by*

$$R(x) = \{v \colon sgn(f(x))(f(x) + \nabla f(x)^\top v) - \epsilon\|\nabla f(x)\| \geq 0\} \cap \{v \colon \|v\|_2 \leq \epsilon\}. \qquad (3)$$

*Proof.* Recall from Definition 1 that the robust set is the set of all directions $v_d$ that the defender $D$ can play at a point $x_i \in \mathcal{X}$ such that no matter what (deterministic) action $v_a$ the attacker $A$ plays, it will always have the utility $-1$, i.e. $u_D(x_i, v_d, v_a) = -1$ for all $v_a \in V$:

$$R(x_i) = \{v \colon v \in V \ s.t. \ \forall v' \in V \ u_A(x_i, v', v) = -1\}$$

Observe that whenever $x_i + v_d$ lies at a distance more than $\epsilon$ from the linear approximation $f_L$ around $x_i$, we have $sgn(f_L(x_i + v_d + v)) = sgn(f_L(x_i + v_d)) \ \forall v \in B(0, \epsilon)$. Hence, no matter what direction $v_a$ the attacker plays, she always gets a utility of $-1$. This shows that $\text{dist}(x_i + v_d, L) \geq \epsilon$ is a sufficient condition for $v_d \in R(x_i)$ given that $v_d$ lies on the same side of $f_L$ as $x_i$, i.e. $sgn f_L(x_i + v_d) = sgn f_L(x_i)$. If $v_d$ lies on the opposite side of $f_L$ as $x_i$, then $v_d \notin R(x_i)$ trivially as the attacker can play $v_a = 0$ to get $sgn f_L(x_i + v_d + v_a) \neq sgn f_L(x_i)$ and thus $+1$ utility.

The above paragraph showing sufficiency of $\text{dist}(x_i + v_d, L) \geq \epsilon$ does not need any restriction on the decision boundary. However, the locally linear model additionally gives us the necessity of the distance condition, as for any $v_d$ played by the defender with $\text{dist}(x_i + v_d, L) < \epsilon$, the attacker can take $v_a$ to be the FGM direction (i.e. perpendicular to $f_L$ towards the other side of the decision boundary as $x + v_d$) to obtain $sgn f_L(x_i + v_d + v_a) \neq sgn f_L(x_i)$, and thus get a $+1$ utility. This shows that $\text{dist}(x_i + v_d, L) \geq \epsilon$ is a necessary condition for $v_d \in R(x_i)$. We have thus shown that the following condition is neccessary and sufficient for $v_d$ to belong to the robust set $R(x_i)$:

(1) The perturbed point staying at least $\epsilon$ away from the boundary, i.e. $\text{dist}(x_i + v, L) \geq \epsilon$ AND
(2) the label staying unchanged, i.e. $sgn f_L(x_i + v) = sgn f_L(x_i)$

The geometry of the problem is shown in Fig. 3. As $\text{dist}(x_i + v, L) = \frac{|f_L(x_i + v)|}{\|\nabla f(x_i)\|_2}$, the first condition gives us $|f_L(x_i + v)| - \epsilon\|\nabla f(x_i)\|_2 \geq 0$. The second condition gives us $|f_L(x_i + v)| = sgn(f(x_i))f_L(x_i + v)$. Since $f_L(x_i + v) = f(x_i) + \nabla f(x_i)^\top v$, we get the equivalent condition:

$$sgn(f(x_i))(f(x_i) + \nabla f(x_i)^\top v) - \epsilon\|\nabla f(x_i)\| \geq 0 \qquad \square$$

The above proof can also be expressed in short by tailoring all parts to our locally linear approximation:

$$R(x) = \{v \in B(0, \epsilon) \colon sgn(f(x))(f(x) + \nabla f(x)^\top (v + v_a)) \geq 0 \ \forall v_a \in B(0, \epsilon)\}$$
$$= \{v \in B(0, \epsilon) \colon \min_{v_a \in B(0, \epsilon)} sgn(f(x))(f(x) + \nabla f(x)^\top (v + v_a)) \geq 0\}$$
$$= \{v \in B(0, \epsilon) \colon sgn(f(x))(f(x) + \nabla f(x)^\top v) - \epsilon\|\nabla f(x)\| \geq 0\}$$

# B  Proof of Lemma 2

**Lemma 2 (FGM is a best-response to any defense).** *For any strategy $s_D \in \mathcal{P}(\mathcal{A}_D)$ played by the defender $D$, the strategy $s_{FGM} \in \mathcal{P}(\mathcal{A}_A)$ played by the attacker $A$ achieves the largest possible utility against $s_D$, i.e., $\bar{u}_A(s_{FGM}, s_D) \geq \bar{u}_A(s_A, s_D)$ for all $s_A \in \mathcal{P}(\mathcal{A}_A)$.*

*Proof.* Let the (possibly randomized) strategies played by the attacker $A$ and the defender $D$ be $s_A$ and $s_D$ respectively. The utility obtained by the attacker is $\bar{u}_A(s_A, s_D)$:

$$\bar{u}_A(s_A, s_D) = \mathop{\mathbb{E}}_{x \sim p_X, a \sim s_A, d \sim s_D} u_A(x, a(x), d(x)) \tag{21}$$

$$= \mathop{\mathbb{E}}_{x \sim p_X, d \sim s_D} \mathop{\mathbb{E}}_{a \sim s_A} \left[ u_A(x, a(x), d(x)) \Big| x, d \right] \tag{22}$$

$$= \int_X \int_{\mathcal{A}_D} \left( \int_{\mathcal{A}_A} u_A(x, a(x), d(x))\, p(a)\, da \right) p(d)\, p(x)\, dd\, dx \tag{23}$$

$$\leq \mathop{\mathbb{E}}_{x \sim p_X, d \sim s_D} \sup_{a \in \mathcal{A}_A} u_A(x, a(x), d(x)) \quad \text{(Property of convex combination)} \tag{24}$$

In the above, we have used the fact that $p(a, d, x)$ factorizes as $p(a)p(d)p(x)$, due to the way our game is played. Recall that for each $x \in X$, we are taking the approximate decision boundary to be the zero-contour of a linear approximation of $f$ around $x$, i.e. $f_L(x') = f(x) + \nabla f(x)^\top (x' - x)$. Additionally, by the definition of $R(x)$, we have the following for all $x \in X, d \in \mathcal{P}(\mathcal{A}_D)$:

$$\sup_{a \in \mathcal{A}_A} u_A(x, a(x), d(x)) = \begin{cases} -1 & \text{if } d(x) \in R(x) \\ +1 & \text{otherwise} \end{cases} \tag{25}$$

Taking the underlying sample-space to be $\Omega = X \times \mathcal{A}_D$, and the associated joint probability distribution over this space to be $p_X \times s_D$, we define the event $E = \{(x, d) : (x, d) \in \Omega,\ d(x) \in R(x)\}$. $\bar{E}$ is defined to be the complement of $E$. From Eq. (25), we see that:

$$\mathop{\mathbb{E}}_{x \sim p_X, d \sim s_D} \sup_{a \in \mathcal{A}_A} u_A(x, a(x), d(x)) = \Pr[\bar{E}] - \Pr[E] \tag{26}$$

Now, we will use simple geometry to see that the FGM direction always achieves the upper-bound obtained in Eq. (24). Recall that the FGM strategy is a deterministic strategy, which plays the funtion $a_{FGM}$ with probability 1 such that the distribution induced on $(X, V)$ has its entire mass on $a_{FGM}(x) = -\epsilon \frac{\text{sgn}(f(x))}{\|\nabla f(x)\|_2} \nabla f(x)$ for all $x \in X$. Following the same steps as above till Eq. (23), we have:

$$\bar{u}_A(s_{FGM}, s_D) = \int_X \int_{\mathcal{A}_D} \left( \int_{\mathcal{A}_A} u_A(x, a(x), d(x))\, p(a|d, x)\, da \right) p(d)\, p(x)\, dd\, dx$$

$$= \int_X \int_{\mathcal{A}_D} u_A(x, a_{FGM}(x), d(x))\, p(d)\, p(x)\, dd\, dx \tag{27}$$

When $d(x) \in R(x)$, all attacker directions lead to an utility of $-1$ for the attacker, hence so does the FGM direction, i.e. $u_A(x, a_{FGM}(x), d(x)) = -1$.

When $d(x) \notin R(x)$, then we claim that $a_{FGM}(x)$ is a direction such that $\text{sgn} f_L(x + d(x) + a_{FGM}(x)) \neq \text{sgn} f_L(x))$. This can be seen by observing that the point closest to $x + d(x)$ on the decision boundary is the first boundary point we hit by moving towards the boundary in a direction perpendicular to it. For the assumed linear boundary $f_L(x) = 0$, this direction is given by $\frac{-\text{sgn}(f(x))}{\|\nabla f(x)\|_2} \nabla f(x)$. Since $d(x) \notin R(x)$, there is atleast one vector $v_a \in V$ such that $\text{sgn} f(x + d(x) + v_a) \neq \text{sgn} f(x + d(x))$. This implies that one can rotate $v_a$ towards the ray $\{x + k \cdot \frac{-\text{sgn}(f(x))}{\|\nabla f(x)\|_2} \nabla f(x) : k \geq 0\}$ to maintain the sign difference. Since the vector obtained on completing this rotation is exactly the FGM direction $a_{FGM}(x)$, we are done. Hence, we have shown that when $d(x) \notin R(x)$, then $u_A(x, a_{FGM}(x), d(x)) = +1$.

Continuing from Eq. (27), we have:

$$\int_X \int_{\mathcal{A}_D} u_A(x, a_{\text{FGM}}(x), d(x)) \, p(d) \, p(x) \, dd \, dx$$

$$= \int_X \int_{\mathcal{A}_D} \Big( (+1) \cdot \mathbb{I}[d(x) \in R(x)] + (-1) \cdot \mathbb{I}[d(x) \notin R(x)] \Big) p(d) \, p(x) \, dd \, dx \tag{28}$$

$$= \Pr[\bar{E}] - \Pr[E] \tag{29}$$

This shows that for all strategies $s_A \in \mathcal{P}(\mathcal{A}_A), s_D \in \mathcal{P}(\mathcal{A}_D)$ played by $A$, $D$ respectively, we have:

$$\bar{u}_A(s_{\text{FGM}}, s_D) \geq \bar{u}_A(s_A, s_D) \qquad\qquad \square$$

## C  Proof of Lemma 3

**Lemma 3 (Randomized Smoothing is a best-response to FGM).** *The strategy $s_{SMOOTH}$ achieves the largest possible utility for the defender against the attack $s_{FGM}$ played by the attacker, i.e., for any defense $s_D \in \mathcal{P}(\mathcal{A}_D)$ we have $\bar{u}_D(s_{FGM}, s_{SMOOTH}) \geq \bar{u}_D(s_{FGM}, s_D)$.*

*Proof.* Let $s_D$ be the strategy followed by $D$, specified by the distribution $p_D \in \mathcal{P}(\mathcal{A}_D)$. Recall that the attackers strategy $s_{\text{FGM}}$ plays the function $a_{\text{FGM}}$ with probability 1 such that the distribution induced on $(X, V)$ has its entire mass on $a_{\text{FGM}}(x) = -\epsilon \frac{\text{sgn}(f(x))}{\|\nabla f(x)\|_2} \nabla f(x)$ for all $x \in X$. The defender's utility can be written as follows:

$$\bar{u}_D(s_{\text{FGM}}, s_D) = \mathbb{E}_{x \sim p_X, d \sim p_D} u_D(x, a_{\text{FGM}}(x), d(x)) \tag{30}$$

$$= \int_X \int_{\mathcal{A}_D} u_D(x, a_{\text{FGM}}(x), d(x)) \, p_D(d) \, p_X(x) \, dd \, dx \tag{31}$$

Following the proof of Lemma 2, we can see that under the FGM attack, the defender gets a utility of $+1$ at the point $x \in X$ when he plays from the robust-set $R(x)$, i.e. for a sample $d \sim p_D$ we have $d(x) \in R(x)$, and a utility of $-1$ otherwise. Accordingly, we now first split the domain in Eq. (31) into two parts depending on whether the defender plays a direction in the robust-set:

$$\bar{u}_D(s_{\text{FGM}}, s_D) = \int_X \int_{\mathcal{A}_D} u_D(x, a_{\text{FGM}}(x), d(x)) \mathbb{I}[d(x) \in R(x)] \, p_D(d) \, p_X(x) \, dd \, dx +$$

$$\int_X \int_{\mathcal{A}_D} u_D(x, a_{\text{FGM}}(x), d(x)) \mathbb{I}[d(x) \notin R(x)] \, p_D(d) \, p_X(x) \, dd \, dx \tag{32}$$

$$R(x) = \int_X \int_{\mathcal{A}_D} (+1) \mathbb{I}[d(x) \in R(x)] \, p_D(d) \, p_X(x) \, dd \, dx +$$

$$\int_X \int_{\mathcal{A}_D} (-1) \mathbb{I}[d(x) \notin R(x)] \, p_D(d) \, p_X(x) \, dd \, dx \tag{33}$$

$$= \int_{\mathcal{A}_D} \int_X (+1) \mathbb{I}[d(x) \in R(x)] \, p_D(d) \, p_X(x) \, dx \, dd +$$

$$\int_{\mathcal{A}_D} \int_X (-1) \mathbb{I}[d(x) \notin R(x)] \, p_D(d) \, p_X(x) \, dx \, dd \tag{34}$$

The order of integration could be interchanged in Eq. (34) since the double integral of the absolute value of the integrand is finite (Fubini's Theorem). We now appeal to the structure of $\mathcal{A}_D$, and recall that $\mathcal{A}_D$ consists of all constant functions from $X$ to $V$. For a particular element $d \in \mathcal{A}_D$, let $v_d$ be its output such that $\forall x \in X \; d(x) = v_d$. Further, we note from the definition of $\phi(v)$ that

$\int_X \mathbb{I}[v \notin R(x)] p_X(x) dx = 1 - \phi(v)$. Continuing from Eq. (34):

$$\bar{u}_D(s_{\text{FGM}}, s_D) = \int_{\mathcal{A}_D} \phi(v_d) \, p_D(d) \, dd - \int_{\mathcal{A}_D} \int_X \mathbb{I}[d(x) \notin R(x)] \, p_D(d) \, p_X(x) \, dx \, dd \qquad (35)$$

$$= \int_{\mathcal{A}_D} \phi(v_d) \, p_D(d) \, dd - \int_{\mathcal{A}_D} (1 - \phi(v_d)) \, p_D(d) \, dd \qquad (36)$$

$$= \int_{\mathcal{A}_D} (2\phi(v_d) - 1) \, p_D(d) \, dd \qquad (37)$$

$$\leq 2\phi(v^*) - 1 \quad \text{(By property of convex combination for any } v^* \in V^*) \qquad (38)$$

Finally, we observe that $s_{\text{SMOOTH}}$ achieves the upper-bound obtained in Eq. (38). Let $p$ be the density for the uniform distribution over the set $F^*$. Following the same steps as above till Eq. (38), we will get the following:

$$\bar{u}_D(s_{\text{FGM}}, s_{\text{SMOOTH}}) = \int_{\mathcal{A}_D} (2\phi(v_d) - 1) \, p(d) \, dd \qquad (39)$$

$$= \int_{F^*} (2\phi(v_d) - 1) \, p(d) \, dd \qquad (40)$$

$$= (2\phi(v^*) - 1) \int_{F^*} p(d) \, dd \qquad (41)$$

$$= (2\phi(v^*) - 1) \qquad (42)$$

Hence, we have have shown that $u_D(s_{\text{FGM}}, s_{\text{SMOOTH}}) \geq u_D(s_{\text{FGM}}, s_D)$ for all $s_D \in \mathcal{P}(\mathcal{A}_D)$. $\qquad \square$

## D  Details for Experiments

Table 2: Attacks and Defenses for MNIST 0 vs 1. The FGM attack and SMOOTH defense correspond to $s_{\text{FGM}}$ and $s_{\text{SMOOTH}}$ respectively. The PGD attack [22] is an iterated version of FGM. True Accuracy shows accuracies using the true classifier $f$ and Approximate Accuracy shows accuracies according to the locally linear approximation $f_L$. Detailed descriptions can be found in the Appendix.

| Attack | Defense | True Accuracy (%) | Approximate Accuracy (%) |
|--------|---------|-------------------|--------------------------|
| - | - | 99.9 | 99.9 |
| FGM | - | 63.0 | 48.3 |
| FGM | SMOOTH | 95.6 | 94.5 |
| PGD | - | 47.7 | 85.6 |
| PGD | SMOOTH | 75.1 | 99.1 |

Table 2 shows results for a particular binary classification task (0 vs 1) on the MNIST dataset. The Table 1 in the main text reports summary statistics for this table over all possible pairs on MNIST and FMNIST. A particular peculiarity is that the network when attacked with PGD has a better *approximate accuracy* than the network attacked with FGM, whereas we know that PGD is a stronger attack than FGM. This happens due to the fact that approximate accuracy is defined on a linearized model (see description in Table 3), whereas PGD observes gradients for the original model, leading to a bad attack performance when the evaluation is made according to the linearized model.

Table 3 is an expanded version of Table 2, where the Description column has been added which shows how the accuracies have been computed. $v_n^*$ is obtained by following the optimization procedure mentioned in Section 4. $a_{\text{PGD}}(x_i)$ is obtained by repeatedly applying FGM and projecting to the set of allowed perturbations $V$, i.e. we iterate the following steps 10 times for each test point $x_i$ to obtain $p_1, p_2, \ldots, p_{10}$, starting with $p_0 = x_i$:

1. Perturb current iterate $p_j$: $p_j' \leftarrow p_j + a_{\text{FGM}}(p_j)$

2. Project perturbation to $B(x_i, \epsilon)$: $p_{j+1} \leftarrow x_i + \epsilon \frac{p_j' - x_i}{\|p_j' - x_i\|}$

At the end we get $a_{\text{PGD}}(x_i) = p_{10} - x_i$.

Table 3: Attacks and Defenses for MNIST 0 vs 1. This is an expanded version of Table 2.

| Attack | Defense | Accuracy (%) | Description | |
|--------|---------|--------------|-------------|---|
| - | - | 99.9 | $\frac{1}{n}\sum_{i=1}^{n}\mathbf{1}[sgnf(x_i)=y_i]$ | |
| FGM | - | 63.0 | $\frac{1}{n}\sum_{i=1}^{n}\mathbf{1}[sgnf(x_i+v_a)=y_i]$ | True |
| FGM | SMOOTH | 95.6 | $\frac{1}{n}\sum_{i=1}^{n}\mathbf{1}[sgnf(x_i+a_{\mathrm{FGM}}(x_i)+v_n^*)=y_i]$ | |
| PGD | - | 47.7 | $\frac{1}{n}\sum_{i=1}^{n}\mathbf{1}[sgnf(x_i+a_{\mathrm{PGD}}(x_i))=y_i]$ | |
| PGD | SMOOTH | 75.1 | $\frac{1}{n}\sum_{i=1}^{n}\mathbf{1}[sgnf(x_i+a_{\mathrm{PGD}}(x_i)+v_n^*)=y_i]$ | |
| - | - | 99.9 | $\frac{1}{n}\sum_{i=1}^{n}\mathbf{1}[sgnf_L(x_i)=y_i]$ | |
| FGM | - | 48.3 | $\frac{1}{n}\sum_{i=1}^{n}\mathbf{1}[sgnf_L(x_i+a_{\mathrm{FGM}}(x_i))=f_L(x_i)]$ | Approximate |
| FGM | SMOOTH | 94.5 | $\frac{1}{n}\sum_{i=1}^{n}\mathbf{1}[sgnf_L(x_i+a_{\mathrm{FGM}}(x_i)+v_n^*)=f_L(x_i)]$ | |
| PGD | - | 85.6 | $\frac{1}{n}\sum_{i=1}^{n}\mathbf{1}[sgnf_L(x_i+a_{\mathrm{PGD}}(x_i))=f_L(x_i)]$ | |
| PGD | SMOOTH | 99.1 | $\frac{1}{n}\sum_{i=1}^{n}\mathbf{1}[sgnf_L(x_i+a_{\mathrm{PGD}}(x_i)+v_n^*)=f_L(x_i)]$ | |

# E   Validity of Modelling Assumptions

While the assumption of local-linearity might seem strong at first, there is ample empirical evidence of its validity for neural networks. Fig. 4, reproduced from [34] shows the decision boundaries of a CNN trained on CIFAR-10 in a $\epsilon$ neighbourhood of many randomly-selected images, where white denotes the predicted class and other shades denote other classes. It can be seen that, locally, the boundary is approximately linear. This *linearity hypothesis* was proposed in [10] and further explored in [34] (showing empirical evidence) and [25] (linking to the existence of universal adversarial perturbations). Recent studies [18], [29] improve Deep Neural Networks' robustness by promoting local-linearity. Hence, we stress that our modelling assumptions *do* partially hold for modern real-world classifiers, and we are not limited to just linear classifiers.

Figure 4: Church-Window plots for a CNN $f$ reproduced from Fig. 11.2 of [34]. Each plot shows $f(\mathbf{x}+a\mathbf{u}+b\mathbf{v})$ for $a,b\in[-\epsilon,\epsilon]$, where $\mathbf{u}$ is the FGM direction, $\mathbf{v}$ is a random direction orthogonal to $\mathbf{u}$ and $\mathbf{x}$ is a random data-point from CIFAR-10. White denotes the class $f(\mathbf{x})$, and other shades denote other classes.