[Reviews · NeurIPS 2020]

Review 1

Summary and Contributions: The authors provide a game theoretic analysis of the game played between attackers and defenders of a given neural network. Instead of changing the weights, the defender (like the attacker) can only change the input in an additive sense. The paper presents arguments that under these constraints the combination of randomized smoother (for the defender) and fast gradient method (for the attacker) form a Nash equilibrium for this game.

Strengths: * The paper addresses a new game in the context of adversarial examples * To my knowledge the paper provides the first claim on a Nash equilibrium for visual perception

Weaknesses: * The presented claims only hold for a locally linear classifier * The claim that the linearity assumption is only violated close to a set of measure 0 is correct. This set has a codimension of 2 in a space of multi-thousand dimensions. Every point that is epsilon-close to this set lies on the codimensional 2 subset after it is dilated by an epsilon ball. It is not clear why this should not cover a large portion of the image domain. * It is not clear why the presented game is practically relevant. That both players add a perturbation to each sample does not seem to reflect the scenario where a classifier is released and the attacker tries to fool it after its release.

Correctness: The claims with respect to the local linearity are not proven. It appears to be incorrect. As a consequence, the presented work might only be true for a linear classifier.

Clarity: The paper is easy to read.

Relation to Prior Work: I did not miss any relevant prior work.

Reproducibility: Yes

Additional Feedback: Post-rebuttal comments: My two major concerns were with respect to 1) practical relevance and 2) the epsilon-closeness to a co-dimension 2 subset. The practical relevance is not addressed properly. The main problem is not that attacker and defender know of each other, but that both are only allowed to change the image. Somehow it makes more sense to me that the defender can also change the weights as in the classical adversarial training setting. I cannot imagine a scenario were the presented scenario is practically relevant. The epsilon-closeness is not clear. The authors explain in their rebuttal that it is in fact an open research question. I agree with the argument that it is difficult to find a co-dim-2-subset. The paper stated that this claim is still true after an epsilon-dilation. This is hard for me to believe without any proper proof. Therefore, I do not see enough arguments to change my vote and recommend to reject the paper at its current stage.


Review 2

Summary and Contributions: The authors model the problem of perturbation-based attacks and defenses for a given two-class classifier as a zero-sum simultaneous game. The main result shows that, under particular restrictions, the FGM attack and randomized smoothing forms an optimal Nash Equilibrium of the game. When samples are finite, the authors showcase a method to obtain an approximation of the defense strategy at the optimal equilibrium and prove that the approximate solution approaches the true equilibrium strategy quickly as $n$ increases.

Strengths: # The game-theoretic formulation is well defined. Given that the game is a zero-sum game, the nash eq = mix-max eq = Stackelberg eq and thus, the authors can choose to relax the simultaneous play assumptions and assume that the attacker is aware of the defender's strategy. Given that the attack is crafting perturbations on a randomly sampled data-point (from p_X) for each attack instance (or game), the attacker knowing that the defender is playing SMOOTH does not change anything. # Algorithms for approximating the defense strategy at the optimal Nash equilibrium and showcasing the improvement of the approximation quality based on the number of samples make the contribution solid. # The paper is well-written.

Weaknesses: # An obvious drawback is that the theory holds for two-class problems with activation functions that enforce piecewise linear hyperplanes. Further, for particular p_x, hyperplanes may be forced to intersect in a majority of places. The theoretical framework is not applicable in such cases. # In regards to the MNIST experiments, quite a few details are missing. What were the epsilon values (for perturbations)? Why select 0 & 1 as the 2 classes? Are similar improvements seen in the case of any two classes of MNIST? Beyond verification of the game-theoretic framework on MNIST, have the authors considered other datasets like FMNIST, or CIFAR(-2)? Often, the results obtained for MNIST do not generalize well to other datasets (for example, most examples for CIFAR-2 or a particular 2-class subset from MNIST may belong to the regions ignored in the game-model, reducing the effectiveness drastically).

Correctness: # Given that ReLU networks may construct parallel hyperplanes that do not intersect but have one class in between them and another class on the two other sides, there may never exist a robust set for points in this region (if the gap between such parallel hyperplanes is $< 2 * \epsilon$; condition 2 in the proof of lemma 1 would never hold for such points). These data-points are also ignored from consideration in the game, beyond the intersection of hyperplanes. # Why is there $R(x)$ on the LHS of equation 33 and 34? (Guessing this is a mistake.) # The authors write "further additive attacks handcrafted for the particular defenses, eg., Carlini-Wagner [1]." This statement is untrue as the CW method was effective against multiple state-of-the-art defenses.

Clarity: # The paper is well-written. # PGD seems to be more optimal against both the zero-perturbation defense and SMOOTH defense (in conflict with Lemma 2 which states that FGM is the optimal attack against any/no defense in this setting). Is that because PGD breaks the assumptions made in the threat model for the attacker? (Also, how does the finite number of samples play into this comparison between PGD and FGM?)

Relation to Prior Work: There exists work on using mixed strategies for using various classifiers against adversarial input in the context of neural network attacks and defenses [1]. In [1], the authors also discuss the additional aspect of losing out on accuracy on test inputs (although for a >2 class classification problem). One question is does the proposed framework put sufficiently high probability to no-defense perturbation ($\epsilon_D = 0$) in areas of the space where using defenses can degrade classification accuracy on benign test inputs? [1] Sengupta, Sailik, Tathagata Chakraborti, and Subbarao Kambhampati. "MTDeep: Moving Target Defense to Boost the Security of Deep Neural Nets Against Adversarial Attacks." Proc. GameSec (2019).

Reproducibility: No

Additional Feedback: I thank the authors for the rebuttal. I wanted to make a few points. First, to respectfully oppose some of the other reviewers, I believe the game-theoretic model is not that impractical. The authors model this as a simultaneous game and the defender can choose some form of input manipulation strategy before passing the input to the classifier. While adversarial training has been the currency in the adversarial defense domain, I don't think this defense model should be stated as impractical and completely ignored by the community. Second, I had also expressed my concerns about the limitations of the assumptions made w.r.t. the epsilon-closeness aspect that others also point out. In this case, I agree with him that without proper proof, it is hard to believe the statement that finding co-dim-2-subset remains difficult after epsilon dilation. On similar lines, I am not sure if the authors can pin-point the assumption that results in the loss of accuracy as pointed out by other reviewers. While the rebuttal doesn't answer all the questions put forth, I want to argue a that this does seem like a decent first step at formalizing the interaction between a particular attack threat model and a particular defense model as a game. I also feel that the authors made an honest effort to provide a PAC style bound (I would like R4 to provide the authors more detail about what their problem with the proof is). I will stick with my initial decision.


Review 3

Summary and Contributions: This paper analyzes the adversarial attack and defense with a two-player zero-sum game framework. It shows that under some assumptions on the model decision boundary (i.e., the decision boundary is locally linear), the Fast Gradient Method attack and the Randomized Smoothing defense form a Nash Equilibrium. It further shows that instead of using a symmetric Gaussian distribution in randomized smoothing, one should select a smoothing distribution based on the classifier. The paper proposes an optimization based method to approximate the defense with a finite training set, and conducts some small scale experiments on the MNIST dataset.

Strengths: The paper rigorously defines a theoretical framework and obtained some interesting insights under this framework and their assumptions. Although this is a theoretically oriented paper, the authors also provide a concrete algorithm that at least can work for a small dataset.

Weaknesses: The assumptions are not reasonable. First, the authors use the locally linear approximation of the model instead of the true decision boundary in defining the utility function. However, in practice these two can be quite different. For example, in the Table 1, the true accuracy and the approximate accuracy are quite different (up to 37.9 absolute percentage). Second, the authors restrict the deterministic defense to always adding a same vector (i.e., agnostic of the test data point). However, in many preprocessing defenses, such as median filter and many other denoising techniques, the defense will depend on the test data point.

Correctness: The claims in the abstract sounds much more exciting than their real findings. FGM is the best strategy under the locally linear approximation model is not that surprising because FGM is designed as the optimal method in the linear case. Most other attacks are designed to overcome the reality that DNNs are highly non-linear. This again can be observed from Table 1 that PGD is much stronger than FGM according to the true accuracy. And by saying that "Randomized Smoothing" is the best defense, most people will interpret it as adding an isometric Gaussian distribution as used in prior work. However, it is not the case in their finding, so it is a bit misleading.

Clarity: The paper is reasonably well written but with a few typos.

Relation to Prior Work: The paper clearly discusses the related work and their main contributions.

Reproducibility: Yes

Additional Feedback: Update after rebuttal: The authors did not fully address my concerns. In particular, the gap between the true accuracy and the approximate accuracy is quite large, and so the utility function used in the paper may not be that meaningful. Therefore, I will keep my score and incline to reject the paper. I suggest that the authors slightly discuss the time complexity and scalability of the methods in section 4. If it is scalable, I suggest doing experiments on a more general dataset, such as (binary version) CIFAR-10.


Review 4

Summary and Contributions: This paper presents a game-theoretic model to analyze the interactions between a defender, who wants to protect the integrity of a classification task, and an attacker, who intends to tamper with the data and minimize the accuracy of the classification. The authors analyzed the equilibrium of the game between the defender and the attacker, demonstrating that a Nash equilibrium is formed when the attacker plays according to FGM and the defender plays according to randomised smoothing. In the case the defender does not know the underlying distribution of the data, it was proposed that the defender can use Randomized Smoothing based on sampled data, and the authors further showed that this approach is a good approximation of the optimal defense strategy.

Strengths: The model is clean, with a simple but representative binary classification task at the core of it. Overall, this is a valid problem to study that is relevant to the NeurIPS community, and the contribution is novel.

Weaknesses: 1. I'm not convinced about the assumption of the defender's strategy, that the same strategy is applied on all the data points. The authors argued that one reason is that the defender does not know the true label of the points, but that doesn't seem to prevent the defender from being able to apply different strategies depending on the relative position of the point to the classification boundary, as well as the distribution P_X. 2. In Section 4, the optimisation problem seems to be a hard one (to maximize the number of feasible linear constraints is NP-hard in general), I'm not sure how efficiently the proposed approach can solve the problem. 3. In Section 5, the convergence of the expected value seems unsurprising. It would be more desired to have some results about how likely the estimated solution would be close to the optimal one (like PAC-learning-style argument).

Correctness: The results look correct to me.

Clarity: The paper is well written and the technical presentation is clear.

Relation to Prior Work: The authors have discussed the related literature and the difference of their work from previous one.

Reproducibility: Yes

Additional Feedback: Please see the Weaknesses section of my comments. =============== Post rebuttal: I thank the authors for their feedback. Regarding the authors' feedback about the PAC style argument, I'm not very convinced. I can understand that E[α] ≤ β implies that Pr[|α− β| > ε] ≤ exp(−2nε^2), but do not understand why Pr[|φ(v^∗) − φ(v_n^∗ ) − β| > ε] ≤ exp(−2nε^2) is also implied when we are given φ(v^∗) − φ(v_n^∗) ≤ α in addition. For example, if φ(v^∗) − φ(v_n^∗ ) = 0 with probability 1, the inequality we want would become Pr[|0 − β| > ε] ≤ exp(−2nε^2), i.e., Pr[|β| > ε] ≤ exp(−2nε^2), which cannot hold true for arbitrary ε if β \neq 0). The authors may want to make this clearer in their paper.

[Author Response · NeurIPS 2020]

We thank the reviewers for finding our problem relevant (R4), the game-theoretic formulation of adversarial attacks and
defenses novel (R1, R3, R4) and the paper clearly written (R1, R2, R3, R4). The main critique is that the assumptions
(local-linearity, restrictions on defender strategies) are strong. However, the reviewers did not point to prior work that is
able to handle our setting. We believe our solution under these assumptions is a necessary stepping stone which lays the
foundations for future theoretical advancements into additive attacks and defenses. We now provide detailed responses:

**[R1, R2, R3, R4] Validity of locally-linear assumption.** While the as-
sumption of local-linearity might seem strong at first, there is ample empir-
ical evidence of its validity for neural networks. Fig. A shows the decision
boundaries of a CNN trained on CIFAR-10 in a $\epsilon$ neighbourhood of many
randomly-selected images, where white denotes the predicted class and
other shades denote other classes. It can be seen that, locally, the boundary
is approximately linear. This *linearity hypothesis* was proposed in [C1] and
further explored in [C2] (showing empirical evidence) and [24] (linking
to the existence of universal adversarial perturbations). Recent studies
[C3], [C4] improve Deep Neural Networks' robustness by promoting local-
linearity. Hence, we stress that our work *does* partially apply to modern
real-world classifiers, and is certainly not limited to linear classifiers. [C1]:
Goodfellow et. al. Explaining and harnessing adversarial examples. [C2]:
Warde-Farley et. al. Adversarial Perturbations of Deep Neural Networks.
[C3]: Lee et. al. Towards Robust, Locally Linear Deep Networks. [C4]:
Qin et. al. Adversarial Robustness through Local Linearization.

Figure A: Church-Window plots for a CNN $f$ reproduced from Fig. 11.2 of [C2]. Each plot shows $f(\mathbf{x} + a\mathbf{u} + b\mathbf{v})$ for $a, b \in [-\epsilon, \epsilon]$, where $\mathbf{u}$ is the FGM direction, $\mathbf{v}$ is a random direction orthogonal to $\mathbf{u}$ and $\mathbf{x}$ is a random data-point from CIFAR-10. White denotes the class $f(\mathbf{x})$, and other shades denote other classes.

**[R1, R2] Do images typically lie near non-linear parts of the decision
boundary?** Yes, empirical evidence on real world networks as in Fig. A
suggests that we almost never find a decision-boundary corner in a $2\epsilon$ neigh-
bourhood around the image. However, these networks partition the input
space into thousands of polyhedra, and the problem of efficiently verifying
whether a given input image lies in a linear-region far from any vertices or
edges of these polyhedra is an open research question [C3].

**[R1] Strong assumption on defender's knowledge.** The design of a game
theoretic framework for analyzing attacks and defenses in a level playing field requires imposing constraints that prevent
the attacker or the defender from always winning. We thus intentionally do not work in a situation where the classifier
is publicly released for attack. Instead, we want to model the reality that the defender will train the classifier knowing
the possible attacks, and in turn the attacker will create new attacks knowing that the publisher was aware of possible
attacks, and so on. This precisely leads to the game-theoretic notion of *perfect knowledge* that we use in Section 2.

**[R3, R4] Strong assumptions on defender's strategies.** As pointed out by R4, obtaining the optimal defense in our
current robust set is already a hard problem. Extending the set to include perturbations dependent on the data-point $x$ will
lead to more complex robust sets, and this is a direction for future work. A first step could be to let $\mathcal{A}_d$ contain all linear
transformations projected to the set of allowed perturbations i.e. $\mathcal{A}_d = \{f_M | f_M \colon \mathcal{X} \to \mathcal{V} \text{ s.t. } f_M(x) = \Pi_\mathcal{V}(Mx)\}$.

**[R3, R4] Scalability of the optimization procedure.** As R4 correctly notes, the optimization problem (7) is hard,
and we present an approximate solution which currently works for small datasets as shown in Table A (it takes $< 10$
seconds for MNIST, FMNIST). In ongoing work, we are scaling it to larger datasets by exploiting the fact that (7) is a
convex-maximization problem and applying techniques from the classical literature on efficient approximations to (7).

**[R2] Extensions to multiple classes, non-ReLU activations, incorpo-
rating test accuracy.** We thank R2 for the suggestions. They are excellent
directions for future work. We will add [Sengupta et. al.] to prior work.

**[R2, R3] Experiments on other datasets.** In our experiments we used
$\epsilon = 4$. Table A shows the result of repeating our experiment in Table 1
of the paper (the column Approximate Accuracy is shown) over all the
55 pairs of classes in MNIST and FMNIST. It can be seen that the trends
observed in the paper hold even when the experiment is repeated over
multiple pairs.

Table A: Mean (Variance) over $\binom{10}{2}$ pairs.

| Attack | Defense | MNIST (%) | FMNIST (%) |
|--------|---------|-----------|------------|
| - | - | 99.9 (0.0) | 99.9 (0.1) |
| FGM | - | 53.3 (10.0) | 47.4 (5.1) |
| FGM | SMOOTH | 71.2 (14.2) | 67.4 (9.0) |
| PGD | - | 71.9 (12.0) | 74.7 (7.3) |
| PGD | SMOOTH | 94.0 (4.0) | 90.3 (8.5) |

**[R2, R3] Minor writing issues** We will fix the typos in Eq. (33), CW method reference, and clarify that we are not
using an isotropic Gaussian for randomized smoothing.

**[R2] Is PGD better than FGM against our defense?** No. Under the locally linear assumption, FGM performs better
than PGD (Table 1: $48.3 < 85.6$ and $94.5 < 99.1$) as expected by Lemma 2. The same trend is seen in Table A.

**[R4] Is there a PAC style argument?** Yes, Sec. 5 establishes a PAC-style bound: The estimated solution is $v_n^*$,
and the optimal one is $v^*$. Eq. (16)-(18) upper bound the difference between the objectives by a quantity $\alpha$, i.e.
$\phi(v^*) - \phi(v_n^*) \leq \alpha$. Eq. (19) establishes that the expectation of $\alpha$ is upper-bounded by a small quantity $\beta$, i.e.
$\mathbb{E}[\alpha] \leq \beta$. Using $\alpha$ in Eq. (10) now yields $\Pr[|\phi(v^*) - \phi(v_n^*) - \beta| > \epsilon] \leq \exp(-2n\epsilon^2)$, which is a PAC bound.

[Meta-Review · NeurIPS 2020]

The paper provides a game-theoretic analysis of additive attacks in the "No-Box" setting. Its most significant result is the proof that the FGM attack and randomized smoothing form a Nash equilibrium under the assumption of a local linearity of the decision boundary. The paper's main contribution is theoretical, its empirical evaluation is performed on the MNIST dataset for a limited number of classes. Also, the validity of some theoretical assumptions is not convincingly presented in the paper. The authors should also clarify the relationship of their work to prior game-theoretic approaches to adversarial learning, e.g., Brückner, M., Kanzow, C. and Scheffer, T., 2012. Static prediction games for adversarial learning problems. The Journal of Machine Learning Research, 13(1), pp.2617-2654. Brückner, M. and Scheffer, T., 2011, August. Stackelberg games for adversarial prediction problems. In Proceedings of the 17th ACM SIGKDD international conference on Knowledge discovery and data mining (pp. 547-555).